# Shifting Treatment Paradigms: Improvements in HR-Positive, HER-2- Negative Breast Cancer Care in Poland from a Clinical Perspective

**DOI:** 10.3390/biomedicines11020510

**Published:** 2023-02-10

**Authors:** Marek Ziobro, Aleksandra Grela-Wojewoda

**Affiliations:** Department of Clinical Oncology, Maria Skłodowska-Curie National Research Institute of Oncology, Kraków Branch, 31-115 Kraków, Poland

**Keywords:** metastatic breast cancer, chemotherapy, endocrine therapy, CDK4/6 inhibitors, ribociclib

## Abstract

Patients with hormone-receptor (HR)-positive, human epidermal growth factor receptor 2 (HER2)-negative breast cancer constitute about 70% of the breast cancer population. About 35% of these patients develop distant metastases and their treatment will be palliative. Cyclin-dependent kinase 4 and 6 (CDK4/6) inhibitors were shown to significantly improve the outcomes of these patients. In combination with endocrine therapy, they have become the standard first-line treatment for HR-positive, HER2-negative breast cancer. In Poland, treatment with CDK4/6 inhibitors is reimbursed only for patients participating in the drug program of the Ministry of Health. However, fulfilling the eligibility criteria for the program may be challenging both for patients and for clinicians. This may lead to a delay in treatment with CDK4/6 inhibitors or a decision to use older and less effective drugs that are more widely available. The aim of this review was to compare the efficacy of first-line therapies in patients with HR-positive, HER2-negative metastatic breast cancer depending on the use of CDK4/6 inhibitors. We compared the efficacy of previous standard therapies with that of ribociclib, a CDK4/6 inhibitor, based on the median progression-free survival (PFS) as an outcome. Median PFS is not affected by the efficacy of subsequent treatment lines and is easy to interpret both for clinicians and for patients. The first-line treatment with chemotherapy or endocrine therapy (without CDK4/6 inhibitors) prolongs median PFS by several months and even to over a dozen months. The first-line treatment with endocrine therapy plus CDK4/6 inhibitors provides an opportunity to achieve a median PFS of more than 25 months and to prolong it by about 9 to 14 months.

## 1. Introduction

The incidence of breast cancer in the Polish population has been increasing over recent years. In 2019, there were 19,620 new cases of breast cancer among women and 149 among men [1].

Currently, in daily clinical practice, breast cancer is classified into two types depending on the expression of the hormone receptor (HR), the human epidermal growth factor receptor 2 (HER2), and the Ki-67 protein in cancer cells. This is sometimes complemented by genetic testing to assess the amplification of the HER2 gene. This approach is a simplified and widely available extrapolation of the classification of breast cancer into subtypes defined by the genetic array testing of the numerous breast cancer genes and then correlating the expression of gene groups with the clinical presentation of cancer and treatment outcomes [2]. Most cases of breast cancer are characterized by HR expression and the absence of HER2 overexpression, referred to as the luminal A or luminal B (HER2 negative) subtype. This cancer phenotype is characterized by a slower progression and requires a different approach to treatment than more aggressive subtypes (such as cancer with HER2 overexpression/amplification or triple negative breast cancer without HR or HER2 expression). Considering the differences in clinical course and treatment, these subtypes of breast cancer were not included in our further analyses.

HR-positive, HER2-negative breast cancer occurs in about 70% of patients with breast cancer worldwide [3]. About 5% to 7% of these patients present with distant metastases at the time of diagnosis. Of the remaining patients receiving radical treatment, about 30% will develop distant metastases in the course of the disease [4,5]. Considering that there are 19,620 new cases of breast cancer annually (based on data from the Polish National Cancer Registry for 2019) [1] and that preinvasive cancer occurs in 7% to 10% of these patients [6], the annual incidence of HR-positive, HER2-negative breast cancer in Poland can be estimated at 12,500 people. Moreover, the number of patients who will develop distant metastases within 1 year is estimated to range from about 4000 to 4500.

For the past 10 years, endocrine therapy has been the treatment of choice for HR-positive, HER2-negative metastatic breast cancer and has been the preferred option even in patients with visceral involvement [7,8,9]. The choice of drugs for the first-line treatment of metastatic breast cancer depends on the type of drugs used as previous adjuvant therapy and a history of primary or secondary resistance to endocrine therapy. However, with advances in the therapeutic methods, the approach to the selection of the optimal treatment regimen has been changing. While 5 years ago, monotherapy with aromatase inhibitors or tamoxifen, or fulvestrant, was the standard first-line treatment [8,9]; today, the treatment of choice is endocrine therapy with aromatase inhibitors or fulvestrant in combination with CDK4/6 inhibitors [10,11]. Current guidelines no longer recommend endocrine therapy alone as an alternative option. Considering differences in the efficacy of these treatments, endocrine therapy with CDK4/6 inhibitors should be the preferred treatment, and it is difficult to identify patients who would derive similar benefits from endocrine therapy alone as the first-line treatment.

Owing to a relatively high cost of treatment with CDK4/6 inhibitors in Poland (vs. previous treatments), the reimbursement is offered only as part of the drug program of the Ministry of Health, “Treatment of breast cancer”. The eligibility criteria and requirements for participation in the program are described in [12]. This approach to reimbursement has had several consequences: it imposes strictly defined requirements for the diagnostic workup and eligibility criteria and limits the availability of treatment to centers that have signed an agreement to implement the program. Treatment guided by the requirements of the drug program allows clinicians to maintain diagnostic and monitoring standards. However, the requirement to perform diagnostic imaging tests prior to treatment can lead to a delay in treatment by several weeks, considering the waiting times for test results. The obligation to strictly follow the schedules for imaging tests constitutes an additional burden for clinicians and patients. Moreover, if the drug program is not available in a given center, the patient has to be transferred to another center to receive treatment with CDK4/6 inhibitors, which usually involves changing the treating physician. In contrast, previous standard treatments, such as endocrine therapy or chemotherapy, can be introduced directly after diagnosing tumor progression, at any cancer treatment center, and the diagnostic workup often carried out at the time the therapy is started. This situation can lead to a delay in treatment with CDK4/6 inhibitors—they may be introduced as a subsequent line of treatment or may not be introduced at all. Obviously, this approach is not in line with current guidelines and knowledge, but considering difficulties in achieving therapeutic goals, clinicians and patients sometimes opt for more easily available options that are associated with less burden in the short term. However, they are often not fully aware of the consequences of such a strategy.

The aim of this review was to compare expected outcomes based on the assessment of median survival times in patients with HR-positive, HER2-negative metastatic breast cancer treated with the widely available types of endocrine therapy alone, chemotherapy alone, and endocrine therapy in combination with CDK4/6 inhibitors. Currently, three CDK4/6 inhibitors are approved for use in these patients and available within the drug program in Poland: ribociclib, palbociclib, and abemaciclib. While there have been no randomized clinical trials comparing these three drugs, meta-analyses and attempts at direct comparisons have shown that they have similar efficacy [13,14]. Therefore, we decided to compare the efficacy of ribociclib [15], as a representative of this drug class, with the efficacy of alternative therapies (not containing CDK4/6 inhibitors) used in the population of patients with HR-positive, HER2-negative metastatic breast cancer. For further simplification, we focused on the effect of therapy on median survival, particularly median progression-free survival (PFS), as an outcome measure that is more intuitive and comprehensible for most oncologists and patients than the widely used hazard coefficients.

The efficacy of cancer treatment depends not only on the type of drugs but also on which line of treatment they are used in. The use of the same drug in the first-line setting is more effective than its use in subsequent lines because cancer resistance to treatment increases over time. Therefore, the eligibility criteria for randomized clinical trials assessing drug efficacy usually clearly determines the line of treatment. These criteria are often transferred from phase III studies to drug programs. An example of such a practice is the “Treatment of breast cancer” program, which currently permits the use of CDK4/6 inhibitors only in the first- or second-line setting [12], according to the summary of product characteristics for ribociclib that permits ribociclib use only after previous endocrine therapy [15]. Considering that randomized clinical trials assessed the use of CDK4/6 inhibitors for first- and second-line treatment and that the drug program also restricts their use to these lines, we also limited our efficacy assessment to these treatment lines and discussed the two settings separately.

## 2. First-Line Treatment for HR-Positive, HER2-Negative Breast Cancer

### 2.1. Endocrine Therapy

Modern endocrine therapy for breast cancer dates back to the 1970s, when tamoxifen (estrogen receptor antagonist) was approved for use and for many years remained the gold standard both as an adjuvant therapy after radical surgery and as a palliative treatment in advanced cancer. In clinical trials, tamoxifen was associated with a median PFS of 5 to 6 months [16,17,18]. Improvement in the efficacy of endocrine therapy was possible following the introduction of third-generation aromatase inhibitors (estrogen receptor degradation inducers) associated with a median PFS of 9 to 14 months (depending on the selected population) [17,18,19,20,21,22]. Further advances in the endocrine therapy of breast cancer were brought about by the approval of fulvestrant (irreversible estrogen receptor blocker). At a dose of 500 mg, fulvestrant improved survival by another few months as compared with aromatase inhibitors [21,22].

The efficacy of the above first-line hormone therapies is compared in Table 1.

The comparison presented in Table 1 shows that the first-line treatment with tamoxifen is associated with a median PFS of 5 to 6 months; aromatase inhibitors with a median PFS of about 9 to 14 months; and 500-mg fulvestrant with a median PFS of about 16 to 23 months. In addition, median PFS tends to increase with the duration of treatment, which is associated with an overall improvement in the quality of maintenance therapy and advances in studies allowing the better classification of patients into individual subtypes of breast cancer. The results summarized in Table 1 are from randomized clinical trials. However, it is important to note that populations treated outside clinical trials usually show slightly worse outcomes due to a higher percentage of patients in a worse clinical conditions.

### 2.2. Chemotherapy

A Cochrane systematic review published in 2003 [7] showed that the first-line endocrine therapy alone and chemotherapy alone had similar efficacy in patients with HR-positive breast cancer. Considering that chemotherapy has higher toxicity, the authors recommended the use of endocrine therapy as the standard first-line treatment in patients with metastatic breast cancer, except in those with visceral crises and rapid tumor progression. An additional argument in favor of endocrine therapy is the improved quality of HR assays and, consequently, a better selection of individual treatments for patients.

At the beginning of the 21st century, the general approach was to consider chemotherapy as subsequent treatment in patients with cancer progression during endocrine therapy, visceral metastases (particularly to the liver), and rapidly progressive disease [7]. Nevertheless, this approach has changed over time, and guidelines developed in 2014 recommended endocrine therapy in these patients due to its lower toxicity and at least comparable efficacy [9]. This, however, does not apply to patients with visceral crisis, that is, liver and lung dysfunction due to rapidly progressive metastatic disease. Therefore, it is difficult to identify clinical data on the efficacy of chemotherapy, especially as the first-line treatment, in the overall population of patients with HR-positive, HER2-negative breast cancer. The MERiDiAN study assessing the efficacy of paclitaxel in combination with bevacizumab and placebo as the first-line treatment reported a median PFS of 9.1 months in the paclitaxel-only arm [23]. Interesting findings were reported in a South Korea study assessing the outcomes of patients with HER2-negative metastatic breast cancer participating in clinical trials between 2002 and 2012 [24]. Patients with HR-positive breast cancer constituted 73.8% of the population. Data on median PFS achieved with first-line chemotherapy using different cytotoxic drugs are shown in Table 2.

In a multivariate analysis, only the presence of HRs was shown to be a significant predictor of longer PFS. According to the authors, this was related to the effect of HR positivity on increased disease control in comparison with the remaining patients with triple negative breast cancer [24]. In summary, the first-line chemotherapy in these patients results in a median PFS of 7.5 to 9 months (with anthracycline- and/or taxane-based regimens).

There are often discrepancies between data from clinical studies and real-life evidence. A large comparative study of first-line treatment in patients with HR-positive, HER2-negative breast cancer was published as part of the Epidemiological Strategy and Medical Economic program [25]. The authors used a national observational database containing data from 18 cancer treatment centers in France, including data on the histological grading of the tumor, the types of treatments, and outcomes. The database contained cases of more than 16,000 women with metastatic breast cancer. The outcomes of different first-line treatments were assessed in a group of 6265 patients with HR-positive, HER2-negative breast cancer, who were potentially sensitive to aromatase inhibitors and who started treatment between 2008 and 2014 [25]. Importantly, patients considered eligible for chemotherapy had a more rapid progression of cancer and had more visceral metastases. On the other hand, patients receiving first-line chemotherapy were younger than those receiving endocrine therapy (median age at diagnosis of metastases was 56 and 66 years, respectively) and were in a better general condition. The outcomes of different first-line treatments are summarized in Table 3.

The authors acknowledged that chemotherapy was administered in patients at higher mortality risk and that maintenance endocrine therapy was in fact proposed only to patients who derived a clinical benefit from chemotherapy [25]. After adjustment for prognostic factors there were no significant differences between patients who received chemotherapy ± maintenance endocrine therapy vs. those who received endocrine therapy alone. A long median PFS reported in this study for endocrine therapy (15.2 months) was explained by patient selection. The authors assessed the data of patients sensitive to aromatase inhibitors, which were administered in 84% of patients receiving endocrine therapy alone and 76% of those receiving chemotherapy. The median PFS of 20.8 months in patients who received chemotherapy and maintenance endocrine therapy was due to the cumulative effect of two treatment lines and patient selection, because maintenance endocrine therapy was proposed to patients who gained a clinical benefit from the first-line chemotherapy [25]. As mentioned above, the reported median PFS for endocrine therapy with aromatase inhibitors ranged from 9 to 14 months (which was also confirmed by Jacquet et al. [25]). Based on this study, it can be concluded that the first-line chemotherapy in patients with HR-positive, HER2-negative breast cancer is associated with a median PFS of 5 to 10 months. These data are in line with previously cited studies [23,24].

According to current European Society for Medical Oncology (ESMO) and American Society of Clinical Oncology (ASCO) guidelines, CDK4/6 inhibitors (cell-cycle inhibitors) are the standard first-line treatment for HR-positive, HER2-negative breast cancer [10,11]. This recommendation is based on data from phase III randomized clinical trials of ribociclib, palbociclib, and abemaciclib that compared standard endocrine therapy with endocrine therapy in combination with a CDK4/6 inhibitor [26,27,28,29,30,31,32]. Data showed a significant improvement in terms of PFS and the percentage of objective responses to treatment, which in some studies translated even to a significant improvement in overall survival (OS). Data on survival times from phase III studies of CDK4/6 inhibitors (for ribociclib) are shown in Table 4.

Data presented in Table 4 for endocrine therapy with a nonsteroidal aromatase inhibitor alone (median PFS, 13.8 and 16.0 months) [26,29] or fulvestrant alone (median PFS, 19.2 months) [27] are in line with the previously cited studies summarized in Table 1 and Table 3 [19,20,21,22,25]. The median PFS in the control arms are in the upper range of results reported for these treatments, which indicates that the studies enrolled patients with favorable prognosis. Despite the satisfactory outcomes for endocrine therapy alone, the results for patients receiving endocrine therapy with ribociclib were significantly better. The addition of ribociclib resulted in a median PFS of 25 to 33 months (unattainable with other types of treatment). Median PFS in patients treated with endocrine therapy plus ribociclib vs. endocrine therapy alone was higher by 9.3, 13.7, and 14.4 months in individual studies. This improvement in outcomes also translated to a significant increase in OS in all three MONALEESA studies, with a median OS longer by about 5 years [30,31,32,33,34].

## 3. Second-Line Treatment of HR-Positive, HER2-Negative Breast Cancer

Compared with the first-line treatment, evidence on the efficacy of the second-line treatment is limited. At the beginning of the 20th century, it was generally believed that patients with disease progression after endocrine therapy should receive chemotherapy and that the efficacy of the first-line chemotherapy was not reduced by previous endocrine therapy. According to the literature [23,25], this would indicate the possibility of achieving a median PFS of 7 to 9 months with chemotherapy. However, Park et al. [24] reported that resistance to different therapies increases with disease progression, and thus the outcomes of chemotherapy after endocrine therapy would be similar to the outcomes of second-line chemotherapy, as shown in Table 5.

Irrespective of the above considerations, a median PFS obtained with second-line chemotherapy ranges from 4 to 9 months [24]. An alternative option to second-line chemotherapy is another endocrine therapy. Based on results from clinical trials, second-line endocrine therapy is associated with a median PFS of 4 to 6 months [35,36,37,38]. Everolimus (an m-TOR inhibitor) with exemestane resulted in a median PFS that was twice as long. However, as everolimus is currently not reimbursed for the treatment of breast cancer in Poland, this option is unavailable for Polish patients. Data on the efficacy of second-line endocrine therapy in HR-positive, HER2-negative breast cancer based on selected phase III studies are presented in Table 6.

In summary, if the first-line treatment of breast cancer fails, the options of endocrine therapy and cytotoxic chemotherapy available outside the drug program can yield a median PFS of 3.5 to 6.5 months (a maximum of 9 months in a more optimistic scenario). The use of CDK4/6 inhibitors, the current gold standard for the first-line treatment of HR-positive, HER2-negative breast cancer [10,11], was also assessed in patients after first-line endocrine therapy failure. In the MONALEESA-3 study [27], a subgroup of patients receiving endocrine therapy as the second-line treatment or who had an early relapse (<12 months after completion of adjuvant therapy) demonstrated a primary resistance to endocrine therapy. The outcomes of the MONALEESA-3 population are presented in Table 7.

The addition of ribociclib to fulvestrant in second-line treatment was associated with a significantly higher PFS and a median PFS of 14.6 months. Compared with the group receiving fulvestrant alone, median PFS was longer by 5.5 months [27,33]. These results support the use of ribociclib as second-line treatment if a CDK4/6 inhibitor was not used before. However, the results for first-line ribociclib are more promising, and studies in the first-line setting included a much larger population of patients [26,28,30,32,33,34].

The above findings are in line with the analysis of outcomes for the subgroup of patients included in the MONALEESA-7 study, presented during the ASCO congress in 2018 [39]. In this study, patients in both treatment arms were classified into subgroups depending on whether they received first-line chemotherapy prior to inclusion in the study. The outcomes of patients included in the MONALEESA-7 study are presented in Table 8.

The median PFS presented in Table 8 for patients after prior chemotherapy is the same as that for second-line endocrine therapy in the MONALEESA-3 study. Moreover, the results for patients not receiving chemotherapy are almost the same as those for the first-line treatment in postmenopausal women in the MONALEESA-2 study. In summary, irrespective of whether the patient receives chemotherapy or endocrine therapy as the first-line treatment, the efficacy of endocrine therapy plus ribociclib as second-line treatment will be significantly lower than that of ribociclib as the first-line treatment. In patients with HR-positive, HER2-negative breast cancer and bone metastases, indications for neurosurgery (if there is a risk of fractures) and palliative radiotherapy should be considered. In these patients, a combination of systemic therapies, especially hormone therapy, plus a CD4/6 inhibitor with local treatment (radiotherapy, neurosurgery) is justified. Such a procedure offers an opportunity to return to physical activity, thus significantly improving the patients’ quality of life. Indications for radiotherapy have not changed in the last few decades. It is the progress in systemic treatment and not radiotherapy that has contributed to the improvement of treatment outcomes in patients with breast cancer.

## 4. Conclusions

In the Polish setting, the treatment of HR-positive, HER2-negative metastatic breast cancer with CDK4/6 inhibitors is currently reimbursed as part of the drug program of the Ministry of Health, and patients can receive this treatment if they fulfill specific criteria. This means that if the CDK4/6 inhibitor is not used as the first-line or second-line palliative endocrine therapy, the patient will not be eligible for reimbursement of this treatment. The first-line treatment with chemotherapy alone or endocrine therapy alone (without CDK4/6 inhibitors) results in a median PFS of a few months up to more than a dozen months. If, instead of endocrine therapy or chemotherapy alone, the patient were to participate in the drug program and were to receive a CDK4/6 inhibitor, then PFS could be prolonged by about 9 to 14 months, resulting in a median PFS of more than 25 months. These outcomes also translate to a significant improvement in OS (based on data from clinical trials). Such a large difference in the efficacy of treatment and the expected benefit of using CDK4/6 inhibitors is a sufficient reason for most patients to justify a delay in treatment by several weeks and to take the effort to fulfill all the eligibility criteria for the drug program. Therefore, clinicians willing to achieve optimal outcomes should preferably treat their patients with first-line endocrine therapy plus a CDK4/6 inhibitor, in line with the current ESMO and ASCO guidelines [10,11]. Treatment without CDK4/6 inhibitors should be limited to the few patients who are not able to fulfill the eligibility criteria for participation in the drug program.

Hormone therapy in combination with CD4/6 inhibitors as the most effective first-line and second-line treatment option for advanced HR-positive, HER2-negative breast cancer (without visceral crisis) should be widely available and should be the preferred therapeutic strategy. Since, in Poland, this treatment can be administered only in cancer centers offering participation in drug programs, its availability is vastly limited. As a result, some patients are treated with suboptimal hormone therapy or less effective and toxic chemotherapy. Another important aspect to consider is that hormone therapy in combination with CD4/6 inhibitors does not compromise the quality of life of patients, which means that professionally and socially active premenopausal women do not have to give up their existing lifestyles. This may bring substantial benefits for individual patients and the economy as a whole. In summary, changes in health regulations in Poland are necessary to optimize the treatment of patients with advanced HR-positive, HER2-negative breast cancer.

## Figures and Tables

**Table 1 biomedicines-11-00510-t001:** Outcomes of first-line hormone therapy based on data for median progression-free and overall survival from the phase III studies of selected drugs used in the endocrine therapy of breast cancer.

Therapy	Median PFS	Median OS	Source/Study	Publication Year
Tamoxifen	5.8	43.3	Paeridens et al. [16]	2008
Tamoxifen	5.6	40	Nabholz et al. [17]	2000
Tamoxifen	6.0	30	Mouridsen et al. [18]	2003
Anastrozole	11.1	39	Nabholz et al. [17]	2002
Anastrozole	10.2	38.2	Bergh et al. [19]	2012
Anastrozole	13.2	41	Metha et al. [20]	2019
Anastrozole	13.8	No data	Robertson et al. [21]	2016
Anastrozole	13.1	48.4	Ellis et al. [22]	2015
Letrozole	9.4	34	Mouridsen et al. [18]	2003
Exemestane	9.9	37.2	Paridaens et al. [16]	2008
Fulvestrant, 500 mg	16.6	No data	Robertson et al. [21]	2016
Fulvestrant, 500 mg	23.4	54.1	Ellis et al. [22]	2015

OS—overall survival; PFS—progression-free survival.

**Table 2 biomedicines-11-00510-t002:** Efficacy of first-line chemotherapy in patients with HER2-negative metastatic breast cancer (HR positivity in 73% of patients) [24].

First-Line Chemotherapy	n	Median PFS (95% CI) (Months)
Whole study group	240	7.6 (6.7–8.5)
Anthracycline-based	85	8.6 (4.9–12.3)
Taxane-based	168	7.7 (6.8–8.6)
Capecitabine-based	90	5.7 (2.3–9.1)

CI—confidence interval; PFS—progression-free survival.

**Table 3 biomedicines-11-00510-t003:** Outcomes of the first-line treatment depending on the type of therapy in French patients with HR-positive, HER2-negative metastatic breast cancer [25].

First-Line Treatment	n	Median PFS (Months)	Median OS (Months)
Whole study group	6265	13.6	53.98
Endocrine therapy AI—84% Antiestrogens—16% Fulvestrant—9% LHRH analogues—9%	2733	15.18	60.7
Chemotherapy ± maintenance endocrine therapy	3532	12.58	49.64
Chemotherapy + maintenance endocrine therapy AI—76% Antiestrogens—28% Fulvestrant—7% LHRH analogues—10%	2073	20.8	62.68
Chemotherapy alone	1459	4.6	30.52

LH-RH—luteinizing hormone-releasing hormone; OS—overall survival; PFS—progression-free survival.

**Table 4 biomedicines-11-00510-t004:** Efficacy of endocrine therapy and endocrine therapy plus ribociclib based on progression-free survival and overall survival in phase III studies [26,28,30,32,33,34].

Therapy		Age	PFS			OS		
	n	Median (Range), Years	Median, Months	Difference, Months	HR (95%CI)	Median, Months	Difference, Months	HR (95%CI)
Letrozole + placebo	334	62 (23–91)	16.0	9.3	0.57 (0.46–0.7)	51.4	12.5	0.76 (0.63–0.93)
Letrozole + ribociclib	334	63 (29–88)	25.3			63.9		
Placebo + fulvestrant	128	63 (31–89)	19.2	14.4	0.55 (0.42–0.72)	51.8	15.8	0.67 (0.50–0.90)
Ribociclib + fulvestrant	237	63 (34–84)	33.6			67.6		
Placebo + goserelin + NSAI	248	43 (25–58)	13.8	13.7	0.569 (0.44–0.74)	48	10.7	0.76 (0.61–0.96)
Ribociclib + goserelin + NSAI	247	43 (29–58)	27.5			58.7		

CI—confidence interval; HR—hazard ratio; OS—overall survival; PFS—progression-free survival; NSAI—nonsteroidal aromatase inhibitors.

**Table 5 biomedicines-11-00510-t005:** Efficacy of second-line chemotherapy in patients with HER2-negative metastatic cancer (HR positivity in 73.8% of patients) [24].

Second-Line Chemotherapy	n	Median PFS (95% CI) Months
Whole study group	209	5.1 (4.3–5.9)
Anthracycline-based	45	5.2 (4.4–6.0)
Taxane-based	47	6.3 (3.3–9.3)
Capecitabine-based	87	5.8 (3.4–8.2)
Gemcitabine/vinorelbine-based	31	4.0 (2.8–5.2)

CI—confidence interval; PFS—progression-free survival.

**Table 6 biomedicines-11-00510-t006:** Second-line endocrine therapy in patients with HR-positive, HER2-negative breast cancer resistant to treatment with nonsteroidal aromatase inhibitors.

Therapy	n	Median PFS/Median TTP, Months	Source/Study
Exemestane	249	3.4	Johnston et al./SoFEA [35]
Fulvestrant	231	4.8	Johnston et al./SoFEA [35]
Fulvestrant + anastrozole	243	4.4	Johnston et al./SoFEA [35]
Exemestane	362	4.1	Baselga et al./BOLERO-2 [36]
Everolimus + exemestane	485	10.6	Baselga et al./BOLERO-2 [36]
Exemestane	351	3.7	Chia et al./EFECT [37]
Fulvestrant	342	3.7	Chia et al./EFECT [37]
Fulvestrant, 250 mg	374	5.5	Di Leo et al./CONFIRM [38]
Fulvestrant, 500 mg	362	6.5	Di Leo et al./CONFIRM [38]

PFS—progression-free survival; TTP—time to progression.

**Table 7 biomedicines-11-00510-t007:** Outcomes of patients receiving second-line endocrine therapy or with an early relapse due to primary resistance to endocrine therapy in the MONALEESA 3 study [27].

Therapy	Progression-Free Survival			Overall Survival		
	Median PFS (Months)	Difference (Months)	HR (95%CI)	Median OS (Months)	Difference (Months)	HR (95%CI)
Placebo + fulvestrant	9.1	5.5	0.57 (0.44–0.74)	32.5	7.7	0.73 (0.53–1.004)
Ribociclib + fulvestrant	14.6			40.2		

CI—confidence interval; HR—hazard ratio; OS—overall survival; PFS—progression-free survival.

**Table 8 biomedicines-11-00510-t008:** Patient outcomes in the MONALEESA-7 depending on whether they received prior first-line chemotherapy: a subgroup analysis [39].

Parameter	After Prior Chemotherapy		Without Prior Chemotherapy	
	Ribociclib n = 47	Placebo n = 47	Ribociclib n = 288	Placebo n = 290
Median PFS (months) (95% CI)	16.6 (10.3–NR)	9.0 (3.9–13.5)	24.7 (19.4–NR)	14.5 (12.2–16.9)
Difference (months)	7.6		10.2	
HR (95% CI)	0.547 (0.314–0.954)		0.566 (0.443–0.724)	

CI—confidence interval; HR—hazard ratio; PFS—progression-free survival; NR—not reached.

## Data Availability

Not applicable.

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
