# Peer review of "Shifting Treatment Paradigms: Improvements in HR-Positive, HER-2- Negative Breast Cancer Care in Poland from a Clinical Perspective"

_biomedicines, 2023, doi:10.3390/biomedicines11020510_

Round 1

Reviewer 1 Report

This is an important study to elucidate the efficacy of first-line therapies in patients with hormone-receptor-positive, human epidermal growth factor receptor2-negative metastatic breast cancer by comparing the application of CDK4/6 inhibitors based on the median progression-free survival in Poland.

In line 49 of page 2, according to reference 3, HR-positive, HER2-negative breast cancer occurs in about 70% of patients with breast cancer in Sweden. Some references worldwide may be added, or the sentence may be revised.

Author Response

We sincerely thank you for the feedback and constructive comments which have led to a substantial improvement of the manuscript. A detailed point-by-point response to your comments is given below.

Reviewer 2 Report

Paper submitted to MDPI Biomedicines, special issue: State-of-the Art Cancer Biology and Therapeutics in Poland, titled: “Overcoming stereotypes in the treatment of advanced HR-positive, HER2-negative breast cancer: clinical practice in Poland” by Marek Ziobro and Aleksandra Grela-Wojewoda compares progress made in a standard clinical practice with treatment of breast cancer patients in Poland.

In general the paper is well organized with information relevant to clinical oncology professionals as well as scientists. However, authors neglect to introduce additional background information, and publication is dependent upon major revisions. The following points need to be addressed prior publication:

1.     Please include references in the main text, while discussing the clinical trials and Tables. The references are inserted in the Tables, but are missing from the text. Please revise the manuscript and correctly insert citations, whenever authors provide data from other studies.

2.     Authors admit that  some of the clinical studies were successful only because of the selection criteria, i.e. patients were selected for specific treatment and thus the treatment response was significant in the pre-selected group. Please provide the details for patient selection for the specific trial mentioned, for a treatment group, as it is important deciding factor for the disease response to the treatment. Pre-selection criteria could be included in the text, while authors refer to the information included in the specific Table. Please be specific, and comment why patient tailored therapy is more effective in increasing patients progression free survival.

3.     In general, Tables that support authors observation include number of patients that the study was conducted on. The Table 4 should also include the average age of patients selected for specific treatment. Table 4 needs to be re-organized.

4.     Please include in a paragraph radiation therapy for HR positive HR-2 negative breast cancer care, especially when combined with CDK4/6 inhibitors.

5.     The drugs studied in clinical trial referred to need to be introduced and characterized from the mechanism of action point of view.

Please in a single sentence categorize the most recently introduced therapeutics and theirs targets and introduce them in the text with the main mechanism of action, such as:

·      CDK4/6 inhibitors, cell cycle inhibitors ribociclib, palbociclib

·      Everolimus, mammalian resistance to rapamycin inhibitor, mTOR inhibitor,

·      Aromatase inhibitors,

·      Letrozole, estrogen receptor degradation promoting drug,

·      Tamoxifen, estrogen receptor antagonist

·      LHRH inhibitor, goserelin used for pre- and peri-menopausal women.

6.     Please consider change of the paper title, for example:

o   Breaking the barriers: recent progress made in HR positive HER-2 negative breast cancer care, clinical perspective from Poland.

o   Changing old to new therapy: improvements made in HR positive HER-2 negative breast cancer care in Poland from clinical perspective.

o   Progress made in HR positive HER-2 negative breast cancer care: from clinical trials to management of breast cancer patients in Poland.

In Conclusions please compare breast cancer patients increased PFS with any data from Polish clinical studies. Please supplement text with potential pitfalls in Breast Cancer care in Poland, and potential solution to changing standards. Comparison between progress made in Western countries and breast cancer care in Poland could be mentioned in a separate section, and include other suggestions, such as laboratory diagnostic research and careful patients selection, in addition to increased funding needed for covering expenses of the new therapeutics in Poland.

Please carefully revise English language, grammar and wording, such as adjunctive, change to adjuvant, gold standard to golden standard ect…

Author Response

(The authors gave the same response as above.)

Reviewer 3 Report

The review article entitled “Overcoming stereotypes in the treatment of advanced HR-positive, HER2-negative breast cancer: clinical practice in Poland” aims to compare expected outcomes based on the assessment of median survival times in patients with HR-positive, HER2-negative metastatic breast cancer treated with the widely available types of endocrine therapy alone, chemotherapy alone, and endocrine therapy in combination with CDK4/6 inhibitors. I consider this paper as reflex of the clinical practice in Poland revising the needs and challenges for breast cancer. The paper is well written and has the makings of a publication. It should include guidelines for future as well as strategies for its implementation.

Author Response

(The authors gave the same response as above.)
